# Identification of distinct loci for de novo DNA methylation by DNMT3A and DNMT3B during mammalian development

Masaki Yagi[1,11,12], Mio Kabata[2,3,12], Akito Tanaka[2], Tomoyo Ukai[1], Sho Ohta[1], Kazuhiko Nakabayashi[4], Masahito Shimizu[3], Kenichiro Hata[4], Alexander Meissner [5,6,7], Takuya Yamamoto [2,8,9,10✉] & Yasuhiro Yamada [1,8✉]

De novo establishment of DNA methylation is accomplished by DNMT3A and DNMT3B. Here, we analyze de novo DNA methylation in mouse embryonic fibroblasts (2i-MEFs) derived from DNA-hypomethylated 2i/L ES cells with genetic ablation of *Dnmt3a* or *Dnmt3b*. We identify 355 and 333 uniquely unmethylated genes in *Dnmt3a* and *Dnmt3b* knockout (KO) 2i-MEFs, respectively. We find that *Dnmt3a* is exclusively required for de novo methylation at both TSS regions and gene bodies of Polycomb group (PcG) target developmental genes, while *Dnmt3b* has a dominant role on the X chromosome. Consistent with this, tissue-specific DNA methylation at PcG target genes is substantially reduced in *Dnmt3a* KO embryos. Finally, we find that human patients with *DNMT3* mutations exhibit reduced DNA methylation at regions that are hypomethylated in *Dnmt3* KO 2i-MEFs. In conclusion, here we report a set of unique de novo DNA methylation target sites for both DNMT3 enzymes during mammalian development that overlap with hypomethylated sites in human patients.

[1] Division of Stem Cell Pathology, Center for Experimental Medicine and Systems Biology, Institute of Medical Science, University of Tokyo, Tokyo 108-8639, Japan. [2] Department of Life Science Frontiers, Center for iPS Cell Research and Application (CiRA), Kyoto University, Kyoto 606-8507, Japan. [3] Department of Gastroenterology/Internal Medicine, Gifu University Graduate School of Medicine, Gifu 501-1194, Japan. [4] Department of Maternal-Fetal Biology, National Research Institute for Child Health and Development, Tokyo 157-8535, Japan. [5] Department of Genome Regulation, Max Planck Institute for Molecular Genetics, Berlin, 14195, Germany. [6] Department of Stem Cell and Regenerative Biology, Harvard University, Cambridge, MA, 02138, USA. [7] Broad Institute of MIT and Harvard, Cambridge, MA, 02142, USA. [8] AMED-CREST, AMED 1-7-1 Otemachi, Tokyo 100-0004, Japan. [9] Institute for the Advanced Study of Human Biology (WPI-ASHBi), Kyoto University, Kyoto 606-8501, Japan. [10] Medical-risk Avoidance based on iPS Cells Team, RIKEN Center for Advanced Intelligence Project (AIP), Kyoto 606-8507, Japan. [11]Present address: Department of Molecular Biology, Massachusetts General Hospital, Boston, MA 02114, USA. [12]These authors contributed equally: Masaki Yagi, Mio Kabata. ✉email: takuya@cira.kyoto-u.ac.jp; yasu@ims.u-tokyo.ac.jp

DNA methylation plays critical roles during mammalian development and ensures stable gene repression, genomic integrity, X chromosome inactivation (XCI), and preservation of genomic imprinting[1–3]. Three conserved DNA methyltransferases, DNMT3A, DNMT3B, and DNMT1, are responsible for de novo establishment and maintenance of DNA methylation patterns[4]. The significance of DNA methyltransferases for mammalian development is highlighted by the fact that genetic deletions of any of these enzymes have lethal phenotypes in mice[5,6]. Moreover, dysregulation of DNMTs is linked to several human diseases, including acute myeloid leukemia (AML) and immunodeficiency, centromere instability, and facial anomalies (ICF) syndrome, indicating that DNA methyltransferases are essential for normal mammalian development and proper maintenance of somatic cells[7,8].

Previous studies revealed the epigenetic mechanisms that establish the strict methylome integrity required for normal development. After epigenetic reprogramming in preimplantation embryos, the DNA methyltransferases DNMT3A and DNMT3B, in conjunction with their nonenzymatic cofactor DNMT3L, mediate global de novo DNA methylation during embryonic and extraembryonic development[5,9,10]. DNMT3A and DNMT3L establish genomic imprinting in gametes[11–13], whereas DNMT3B and DNMT3L are required for unique DNA methylation patterns in placental progenitors[10]. The genomic enrichment patterns of DNMT3A differ from those of DNMT3B in mouse embryonic stem (ES) cells, suggesting that each de novo methyltransferase has unique targets during development[14,15]. Indeed, individual genetic ablation of these methyltransferases results in lethal phenotypes at different developmental stages[5]. However, it remains unclear whether the differential targeting of DNMT3 enzymes is responsible for locus-specific de novo DNA methylation during embryonic development.

To comprehensively identify the target sites for de novo DNA methylation by the DNMT3 enzymes, in the present study, we took advantage of female mouse ES cells established in the presence of MEK and Gsk3 inhibitors, which lack most DNA methylation (2i/L ES cells)[16]. In combination with genetic ablation of Dnmt3a or Dnmt3b, here we report unique de novo DNA methylation target sites for both DNMT3 enzymes during embryonic development.

## Results

### DNA hypomethylated ESCs for de novo methylation analysis.
Given the early embryonic lethal phenotype of Dnmt3b knockout (KO) mice, ES cells could be a powerful tool for tracking and comparing the de novo methylation activity of DNMT3s during embryonic development (Fig. 1a). It should be noted, however, that mouse ES cells cultured in serum and leukemia inhibitory factor (LIF) (S/L ES cells) exhibit global DNA hypermethylation relative to inner cell mass (ICM) cells, which are the in vivo counterpart of ES cells[17]. Indeed, de novo methylated regions at post-implantation epiblasts in vivo are already methylated in S/L ES cells (Fig. 1b, c). Moreover, Polycomb group (PcG) target transcription factor genes, which often include key developmental genes, are highly methylated in S/L ES cells, but still hypomethylated in both ICM and epiblast (Fig. 1d and Supplementary Fig. 1a), precluding the use of S/L ES cells for analysis of de novo methylation at early developmental stages. In a previous study, we showed that mouse ES cells established under 2i/L (MEK inhibitor, Gsk3 inhibitor, and LIF) culture conditions (2i/L ES cells) exhibit a substantial reduction in global DNA methylation levels[16]. Female 2i/L ES cells lack DNA methylation at most sites, including the PcG target genes (Fig. 1b–d and Supplementary Fig. 1a). Collectively, these findings indicated that hypomethylated female

2i/L ES cells represent a powerful tool for de novo methylation during early embryonic development.

In this study, to identify the unique targets of de novo methylation by the DNMT3 enzymes, we used female mouse 2i/L ES cells with targeted deletions in either Dnmt3a or Dnmt3b (Fig. 1a). Dnmt3 KO 2i/L ES cells grow well and are morphologically normal under 2i/L culture conditions[16]. In addition, they efficiently contribute to chimeric embryos as efficiently as wild-type (WT) 2i/L ES cells, indicating that Dnmt3 KO 2i/L ES cells retain pluripotency and the ability to differentiate into somatic cells in vivo[16]. To compare the targets of each DNA methyltransferase, we established mouse embryonic fibroblasts (MEFs) from chimeric mice derived from female Dnmt3 KO 2i/L ES cells (Dnmt3 KO 2i-MEFs), followed by antibiotic selection, and assessed DNA methylation profiles by whole-genome bisulfite sequencing (WGBS) and target-captured MethylC-seq analysis (Supplementary Fig. 1b). Considering that the 5 hmC level is negligible in ESCs[18], we regarded both 5 hmC and 5 mC as 5 mC in this study.

CpG methylation on all chromosomes was lower in both Dnmt3a and Dnmt3b KO 2i-MEFs than in WT 2i-MEFs (Fig. 1e and Supplementary Fig. 1c). Extending the analysis to various genomic features, including transposable elements, we found that both Dnmt3a and Dnmt3b KO decreased DNA methylation levels on all elements (Fig. 1f, g). Consistent with a previous study[19], Dnmt3b KO 2i-MEFs exhibited a clear reduction in DNA methylation on transposable elements relative to WT or Dnmt3a KO 2i-MEFs (Fig. 1g). As shown in our previous study[16], 2i-MEFs exhibited global loss of genomic imprinting regardless of Dnmt3 status (Supplementary Fig. 1d). When we divided all CpG sites into three categories according to DNA methylation levels [hypermethylated (>0.8), intermediate (0.2–0.8), and hypomethylated (<0.2) sites], we found that hypermethylated CpG sites were less abundant in Dnmt3b KO 2i-MEFs, whereas Dnmt3a KO 2i-MEFs exhibited relatively modest reduction (Fig. 1h). The reduced methylation level in Dnmt3b KO 2i-MEFs was observed at regions marked by various histone marks, such as H3K36me3 and H3K9me3 (Supplementary Fig. 2a)[14]. We also found that Dnmt3b KO epiblasts exhibit reduced CpG methylation levels at de novo methylated regions during peri-implantation (from ICM to post-implantation epiblasts) relative to WT or Dnmt3a KO epiblasts (Supplementary Fig. 2b). Together, these results suggest that DNMT3B de novo DNA methylation activity is predominant in most genomic regions during early development.

Consistent with a previous report[20], methylation levels at CGIs on the X chromosome were markedly lower in Dnmt3b KO 2i-MEFs than in Dnmt3a KO 2i-MEFs (Fig. 1i and Supplementary Fig. 2c) (3a KO vs WT: $p = 3.6 \times 10^{-1}$, 3b KO vs WT: $p = 1.2 \times 10^{-15}$; two-sided, Kolmogorov–Smirnov test), suggesting that Dnmt3b is largely responsible for de novo CGI methylation on the X chromosome. Non-CGI sites were also less methylated in Dnmt3b KO 2i-MEFs, which was more pronounced on the X chromosome than on autosomes (Fig. 1i and Supplementary Fig. 2c–e) (3a KO vs WT: $p < 2.2 \times 10^{-16}$, 3b KO vs WT: $p < 2.2 \times 10^{-16}$; two-sided, Kolmogorov–Smirnov test). In sharp contrast, Dnmt3a KO 2i-MEFs did not exhibit an apparent reduction in non-CGI methylation on the X chromosome (Fig. 1i and Supplementary Fig. 2c). Collectively, these results suggest that DNMT3B is the enzyme predominantly responsible for global de novo methylation on the X chromosome during XCI.

### Target genes of de novo methylation by DNMT3A and DNMT3B.
We next sought to identify, in an unbiased manner, unique target loci of DNMT3A- and DNMT3B-mediated de novo DNA methylation during embryonic development. To this end,

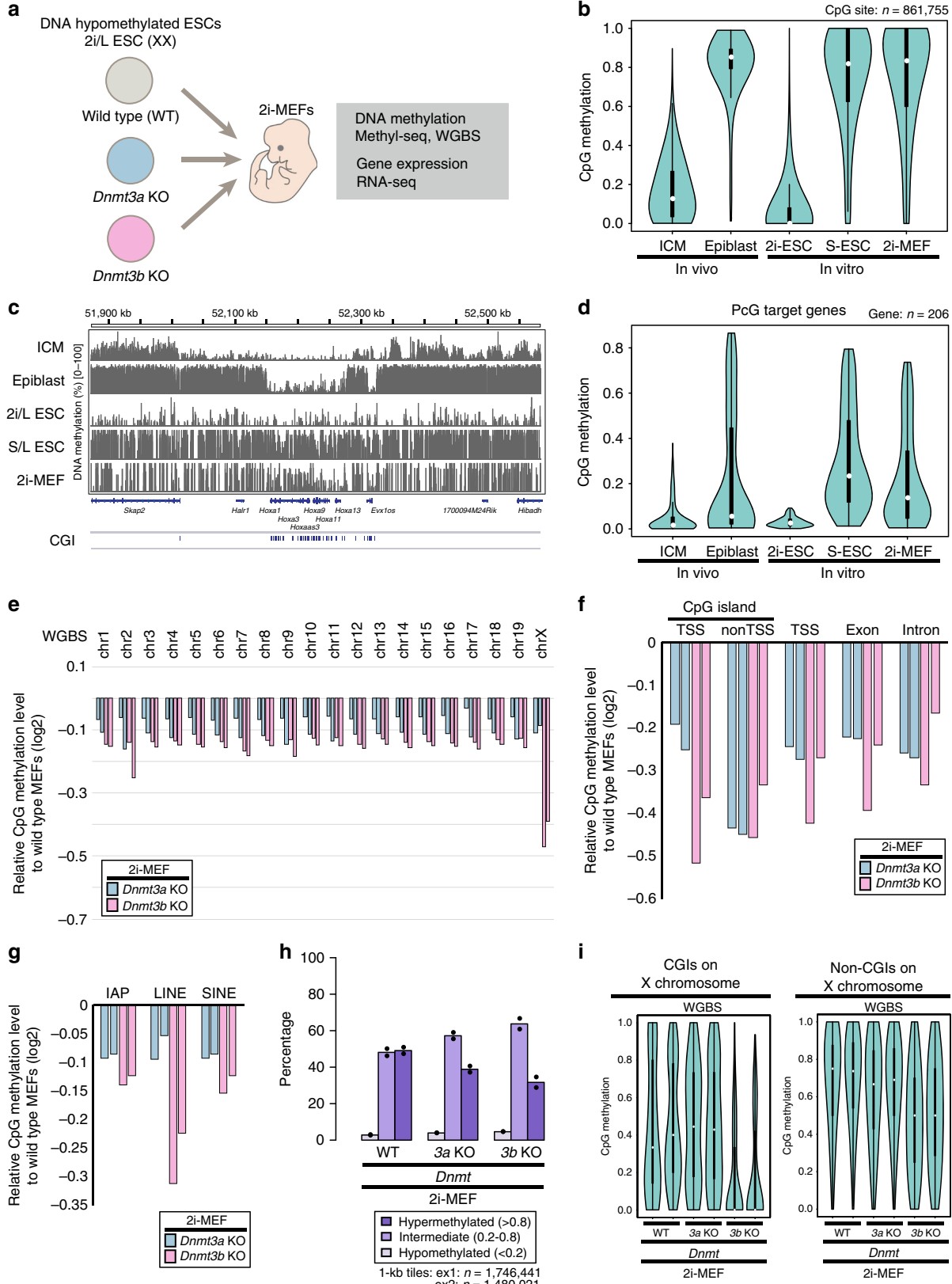

we first identified differentially methylated tiles (DMT; non-overlapping 1k tiles) that differed significantly between WT 2i-MEFs and each KO 2i-MEFs by using WGBS data. Consistent with previous reports[5,21], DNA methylation levels at most regions did not significantly differ between *Dnmt3a* and *Dnmt3b* KO 2i-MEFs, but we identified thousands of DMTs in either of the KO

(4308 and 7357 DMTs in *Dnmt3a* and *Dnmt3b* KO 2i-MEFs, respectively) (Fig. 2a and Supplementary Data 1). A small subset of DMTs (*n* = 143) exhibited a modest reduction in DNA methylation in both *Dnmt3a* KO and *Dnmt3b* KO 2i-MEFs, suggesting that these loci are sufficiently methylated only when DNMT3A and DNMT3B are both present (Fig. 2a and

**Fig. 1 DNA hypomethylated ES cells for de novo methylation analysis. a** Schematic of experimental design. Either *Dnmt3a* or *Dnmt3b* was disrupted by CRISPR/Cas9 in female 2i/L ES cells[16]. **b** CpG methylation levels at loci that were differentially methylated between ICM and epiblast (see the methods section for details) in 2i/L ES cells and S/L ES cells. WGBS data were used for the analysis. White dots indicate median methylation levels. Black bars and the lines stretched from the bar represent IQR and the lower/upper adjacent values (±1.5 IQR), respectively. ICM and epiblast data were obtained from GSE84236[10]. Data of 2i/L ES cells, S/L ES cells, and 2i-MEFs are from GSE84165[16]. **c** CpG methylation levels at a representative genomic region including the *Hoxa* cluster, as determined by WGBS. Each bar indicates a CpG site, and the height of the bar represents methylation percentage (0–100%). Locations of genes and CpG islands (CGIs) are indicated below. Data are from GSE84236[10] and GSE84165[16]. **d** CpG methylation levels at PcG target developmental genes in ICM, epiblast, 2i/L ES cells, S/L ES cells, and 2i-MEFs, as determined by WGBS. Data were obtained from GSE84236[10] and GSE84165[16]. **e** Relative CpG methylation level [log$_2$(fold change)] at each chromosome in *Dnmt3* KO 2i-MEFs vs. wild-type 2i-MEFs, as determined by WGBS. Data from two independent experiments are shown. **f** Relative CpG methylation levels [log$_2$(fold change)] at CGIs, promoters [Transcription Start Site (TSS) ± 1000 bp], exons, and introns in *Dnmt3* KO 2i-MEFs vs. wild-type 2i-MEFs, as determined by MethylC-seq. **g** Relative CpG methylation levels [log$_2$(fold change)] at transposable elements (IAPs, LINEs, and SINEs) in *Dnmt3* KO 2i-MEFs vs. wild-type 2i-MEFs by WGBS. h: Fraction of hypermethylated (>0.8), intermediate (0.2–0.8), and hypomethylated (<0.2) CpG sites in wild-type and each type of KO 2i-MEFs. WGBS data (CpG sites at ≥5× coverage) were used for this analysis. **i** CpG methylation levels at CGIs and non-CGIs on the X chromosome in WT, *Dnmt3a* KO, and *Dnmt3b* KO 2i-MEFs, as determined by WGBS. White dots indicate median methylation levels.

Supplementary Data 1). To uncover the specific target genes of each methyltransferase, we used MethylC-seq data that are enriched and have high coverage reads in gene regions. Notably, we identified 355 and 333 specific target genes of DNMT3A and DNMT3B, respectively (Fig. 2b, c and Supplementary Fig. 3a, b) (Supplementary Data 2–5). Gene ontology (GO) analyses revealed that DNMT3B-specific genes are involved in synaptonemal complex assembly (e.g., *Tex12*, *Syde1*, and *Meioc*), plasma membrane (e.g., *Pcdha/b* family), and cell proliferation in forebrain (e.g., *Wnt7a*, *Wnt7b*, *Wnt3a*, and *Zeb2*) (Supplementary Fig. 3c, and Supplementary Data 4 and 5). Our analysis identified the previously reported DNMT3B-specific targets, *Pcdha/b* family genes[22], confirming that our strategy can successfully identify the target genes of the de novo methyltransferase (Supplementary Fig. 3a). By contrast, DNMT3A target genes included many transcription factors that are essential for normal development, including genes of the *Hox*, *Gata*, and *Cbx* families (Fig. 2c and Supplementary Data 3 and 4). Consistent with this, GO analyses indicated that DNMT3A target genes are involved in sequence-specific DNA binding (e.g., *Nr6a1*, *Gata2*, *Hoxc9*, and *Pou5f1*), transcription (e.g., *Zfp46*, *Barx1*, and *Osr1*), and pharyngeal system development (e.g., *Gata3*, *Six1*, and *Tbx1*) (Supplementary Fig. 3d). Because the DNMT3A target genes included many developmental genes that are also known as PcG target transcription factor genes[23], we next hypothesized that DNMT3A regulates de novo DNA methylation at PcG target developmental genes during embryonic development. Notably, most of the PcG target genes that acquire de novo DNA methylation during embryonic development were strikingly hypomethylated in *Dnmt3a* KO 2i-MEFs (Fig. 2d and Supplementary Fig. 3e) [3a KO vs WT (mean methylation levels): $p = 5.6 \times 10^{-48}$, 3b KO vs WT (mean methylation level): $p = 8.2 \times 10^{-16}$, Wilcoxon signed-rank test]. For example, our data showed that all *Hox* gene clusters (*Hoxa*, *Hoxb*, *Hoxc*, and *Hoxd*) located on different chromosomes, as well as other PcG target transcription factors, were specifically methylated by DNMT3A (Fig. 2c, Supplementary Fig. 3b, and Supplementary Data 3). The reduced DNA methylation in *Dnmt3a* KO 2i-MEFs was observed in both short and long genes (Supplementary Fig. 4a). To confirm DNMT3A-mediated de novo methylation, we next examined the effect of ectopic *Dnmt3a* expression on reduced methylation at PcG target developmental genes in *Dnmt3a* KO cells. In these experiments, we introduced a doxycycline (Dox)-inducible *Dnmt3a1* expression vector into *Dnmt3a* KO 2i/L ES cells, and then differentiated them into epiblast-like cells (EpiLCs) (Supplementary Fig. 4b). Forced expression of *Dnmt3a1* in *Dnmt3a* KO cells increased de novo DNA methylation at *Hoxa1* and *Meis1* to a level similar to that observed in WT cells (Supplementary Fig. 4c, d and Source

Data file). Based on these findings, we concluded that *Dnmt3a* confers de novo DNA methylation at PcG target developmental genes upon ES cell differentiation.

To more thoroughly examine DNMT3A-mediated de novo methylation at PcG target developmental genes, we divided the region of each PcG target gene into the Transcription Start Site (TSS) region (TSS ± 1000 bp) and gene body [TSS to Transcription End Site (TES)] and analyzed the methylation patterns (Fig. 2e). The TSS regions of a subset of PcG target developmental genes exhibited lower levels of de novo methylation in *Dnmt3a* KO 2i-MEFs than in WT 2i-MEFs (Fig. 2e) [3a KO vs WT (mean methylation levels, TSS): $p < 2.2 \times 10^{-16}$, 3b KO vs WT (mean methylation levels, TSS): $p = 5.4 \times 10^{-14}$; two-sided, Wilcoxon signed-rank test]. In contrast to the repressive role of DNA methylation at TSS regions, gene body methylation is generally associated with active transcription[24]. Consistent with this, gene bodies of more highly expressed PcG target developmental genes tended to have higher methylation levels in 2i-MEFs (Fig. 2e). In addition, DNMT3B binds to gene bodies of actively transcribed genes marked by H3K36me3, and is responsible for de novo DNA methylation at the gene bodies[14,25]. Indeed, H3K36me3-marked regions were less methylated in *Dnmt3b* KO 2i-MEFs than in *Dnmt3a* KO 2i-MEFs (Supplementary Fig. 2a). Importantly, however, gene bodies of PcG target genes were less methylated in *Dnmt3a* KO 2i-MEFs than in *Dnmt3b* KO 2i-MEFs (Fig. 2e and Supplementary Fig. 5a) [3a KO vs WT (Gene body): $p < 2.2 \times 10^{-16}$, 3b KO vs WT (Gene body): $p = 8.9 \times 10^{-3}$; two-sided, Wilcoxon signed-rank test]. The reduced methylation at gene bodies in *Dnmt3a* KO 2i-MEFs was similarly observed at those distant from TSS (Supplementary Fig. 5b), indicating that the observed changes are not always associated with close proximity of CpG islands linked to promoters. Collectively, these results suggest that, in sharp contrast to other genes, gene bodies of PcG target developmental genes are preferentially methylated by DNMT3A, consistent with a recent study demonstrating that DNMT3A1, a longer isoform of DNMT3A, preferentially localizes to the methylated shores of bivalent CGIs in ES cells[15]. Given that DNMT3A, together with DNMT3L, establishes genomic imprinting in gametes, we examined the involvement of DNMT3L in DNMT3A-mediated de novo methylation at PcG target genes. However, methylation levels of PcG target genes were comparable between *Dnmt3l* KO and WT E14.5 MEFs (Supplementary Fig. 5c), demonstrating that DNMT3L is dispensable for DNMT3A-mediated methylation at PcG target genes during embryonic development.

**Differential binding patterns of DNMT3A1 and DNMT3B1.**
To obtain mechanistic insights into the locus specificity of

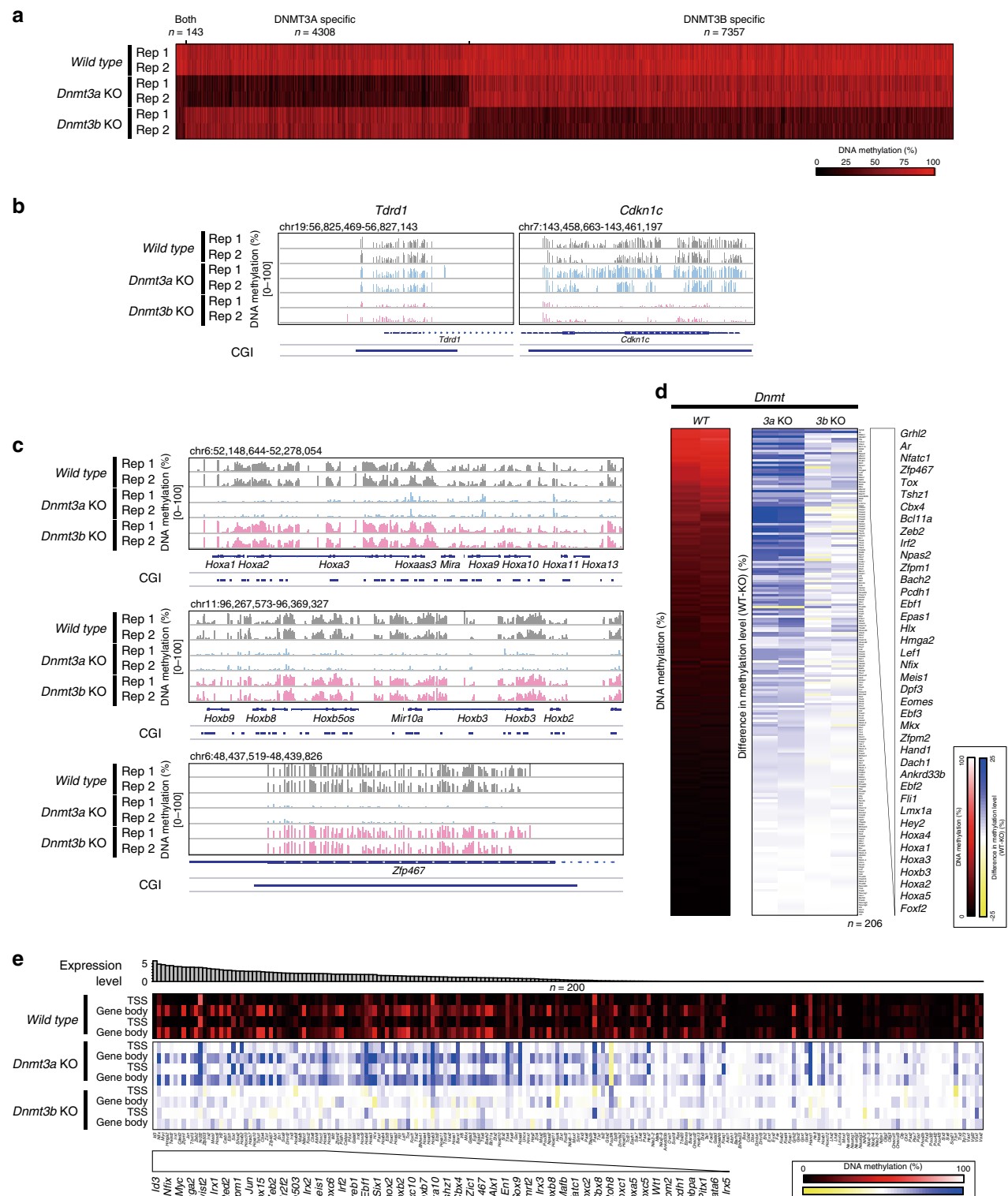

DNMT3 enzymes for de novo methylation, we examined genomic enrichment patterns of DNMT3A and DNMT3B by analyzing ChIP-seq datasets[14,15]. Consistent with previous reports demonstrating that DNMT3B is responsible for gene body methylation of actively transcribed genes[14,25], we confirmed enrichment of DNMT3B1, but not DNMT3A1, in gene bodies (Fig. 3a). Notably, DNMT3A, and especially DNMT3A1, was enriched at DNMT3A target genes (Fig. 3a and Supplementary Fig. 6a). By contrast, enrichment of DNMT3B1 was not altered at

target genes of DNMT3B-mediated de novo methylation (Fig. 3a), suggesting the existence of distinct mechanisms governing the locus-specific activity of de novo methyltransferases. Considering that DNMT3A mediates de novo methylation at PcG target developmental genes, we next analyzed the binding patterns of the DNMT3 enzymes on the PcG target genes. We observed prominent enrichment of DNMT3A1 binding on PcG target developmental genes, including enrichment in gene bodies (Fig. 3b). Remarkably, DNMT3B1 binding was reduced on the

**Fig. 2 Target genes of de novo methylation by DNMT3A and DNMT3B. a** Heatmap of CpG methylation levels within 1 K tiled regions in which CpG methylation levels differed between the WT and each *Dnmt3* KO. Differentially methylated 1 K tiled regions are identified using the Bioconductor DSS package with methylation status of nonoverlapping 1 K tiles that have ≥10 read coverages at CpG sites within a tile across all samples (see the methods section in detail). **b** Representative loci of DNMT3B-specific target sites in 2i-MEFs. Each bar indicates a CpG site, and the height of the bar represents methylation percentage, as determined by MethylC-seq (0–100%). Locations of genes and CGIs are indicated below. **c** Representative loci of DNMT3A-specific target sites in 2i-MEFs. Essential developmental genes are frequently unmethylated in *Dnmt3a* KO 2i-MEFs. Each bar indicates a CpG site, and bar height represents methylation percentage by MethylC-seq (0–100%). Locations of genes and CGIs are indicated below. **d** The left panel shows mean DNA methylation levels at CpG sites within each PcG target developmental gene[23] [from TSS to Transcription End Site (TES)] in WT 2i-MEFs from two independent experiments. The right panel shows methylation difference between WT and each *Dnmt3* KO 2i-MEFs from two independent experiments. Genes are sorted by the mean methylation levels in the WT. Color scales indicate DNA methylation levels and their differences, respectively. **e** The upper panel shows mean DNA methylation levels at CpG sites within TSS ± 1000 bp and gene body (TSS-TES) of each PcG target gene in WT 2i-MEFs from two independent experiments. The lower panel shows methylation difference between WT and each type of *Dnmt3* KO 2i-MEFs from two independent experiments. Color scale indicates DNA methylation levels and their difference, respectively. PcG genes are ordered based on expression levels (FPKM values) in WT 2i-MEFs (top panel).

PcG target genes[23] (Fig. 3b), indicating that DNMT3B1 is precluded from these loci. Among DNMT3A target genes, further enrichment of DNMT3A1 was observed on PcG target developmental genes (Fig. 3c), and reduced binding of DNMT3B1 was detected exclusively at DNMT3A target genes that were also PcG target genes (Fig. 3c), highlighting unique binding patterns of DNMT3 enzymes at PcG target loci. Together, these results suggest that the preferred physical interactions of DNMT3A and DNMT3B are associated with the locus-specific activity of the enzymes at PcG target developmental genes.

**Derepression of PcG target genes in *Dnmt3a* KO 2i-MEFs.** Given that DNA methylation at CGIs linked to TSS regions are associated with gene repression, we next investigated the functional implications of DNMT3-mediated de novo methylation on gene expression during embryonic development. Because *Dnmt3b* KO 2i-MEFs exhibited global CGI hypomethylation at the X chromosome, we first performed RNA-seq analysis to compare global gene expression profiles of X-linked genes between WT and both types of *Dnmt* KO 2i-MEFs. Despite the significant reduction in CGI methylation in *Dnmt3b* KO 2i-MEFs, expression levels of X-linked genes were comparable among WT, *Dnmt3a* KO, and *Dnmt3b* KO (*3a* KO: $p = 7.6 \times 10^{-1}$, *3b* KO: $p = 8.2 \times 10^{-1}$) (Fig. 4a), suggesting that DNA methylation–independent mechanisms ensure dosage compensation of X-linked genes in *Dnmt3b* KO 2i-MEFs[26]. We also examined expression of PcG target developmental genes in WT and *Dnmt* KO 2i-MEFs. Although reduced CGI methylation at PcG target developmental genes was exclusively detectable in *Dnmt3a* KO 2i-MEFs, expression levels of most PcG target developmental genes were not altered in *Dnmt3a* KO 2i-MEFs (*3a* KO: $p = 7.3 \times 10^{-1}$, *3b* KO: $p = 6.5 \times 10^{-1}$) (Fig. 4b). However, when we focused on the TSS regions of PcG target developmental genes expressed at low levels, which were methylated in WT 2i-MEFs (>15%), we found that a small subset of the PcG target genes (e.g., *Pax8*) was modestly derepressed in *Dnmt3a* KO 2i-MEFs, accompanied by loss of DNA methylation at TSS regions (Fig. 4c–e and Supplementary Fig. 6b). To further investigate DNMT3A-mediated repression of PcG target developmental genes, we also obtained MEFs from *Dnmt3a* WT and null embryos by crossing *Dnmt3a* heterozygous KO mice. Importantly, derepression of *Pax8* was confirmed in *Dnmt3a* KO MEFs (Supplementary Fig. 6c). Collectively, these data indicate that DNMT3A-mediated DNA methylation plays a role in stable silencing of a subset of PcG target developmental genes.

**Tissue-specific methylation at PcG target genes by DNMT3A.** Our previous study demonstrated that female 2i/L ES cells exhibit a massive erasure of genomic imprints and harbor an impaired autonomous developmental potential[16], which raised the possibility that the observed changes in DNA methylation are not directly associated with DNMT3 de novo DNA methylation activity. To investigate DNMT3A-mediated de novo methylation during embryonic development in vivo, we obtained various tissues (brain, limb, heart, lung, and liver) from *Dnmt3a* WT and null embryos at E14.5 (Fig. 5a, b). We then performed MethylC-seq analysis to compare DNA methylation patterns at PcG target developmental genes in different tissues (Supplementary Fig. 1b). Notably in this regard, PcG target genes in different tissues exhibited distinct DNA methylation levels in WT embryos. For example, the *Hoxb* cluster is methylated in the heart and liver, but remains unmethylated in the brain (Fig. 5c). Notably, genetic ablation of *Dnmt3a* resulted in a reduction in DNA methylation at the *Hoxb* cluster in the heart and liver (Fig. 5c). Moreover, other PcG target developmental genes such as Chromo, T-box, and Hox families also exhibited tissue-specific methylation patterns in WT organs, but were hypomethylated in *Dnmt3a* KO tissues (Fig. 5d–e and Supplementary Fig. 7). Conventional bisulfite sequencing analysis confirmed that the tissue-specific methylation at *Meis1* is mediated by DNMT3A (Supplementary Fig. 8a). Consistent with the notion that PcG target developmental genes are regulated by cooperative epigenetic mechanisms, most PcG target genes were expressed at similar levels in WT and *Dnmt3a* KO tissues (Fig. 5f and Supplementary Fig. 8b). Notably, gene bodies of transcriptionally active PcG target developmental genes were highly methylated in both the heart and liver, but were less methylated in *Dnmt3a* KO tissues (Supplementary Fig. 8c), affirming that de novo methylation at gene bodies of PcG target developmental genes is mediated by DNMT3A in vivo. Taken together, these results underscore the essential role of DNMT3A in establishing tissue-specific DNA methylation patterns at PcG target developmental genes during embryonic development.

**Reduced DNA methylation in patients with *DNMT3* mutations.** Finally, we investigated the unique functions of DNMT3A and DNMT3B in humans. Somatic inactivating mutations of *DNMT3A* and *DNMT3B* have been reported in AML[27] and ICF syndrome patients[7], respectively. To investigate the possible link between an impaired DNMT3 function and altered DNA methylation in human diseases, we first analyzed DNA methylation status at mouse DNMT3B target genes in ICF patients harboring a *DNMT3B* mutation. Human homologs of a subset of mouse DNMT3B target genes were similarly hypomethylated in *DNMT3B* mutant ICF patients relative to control patients (Fig. 6a). Importantly, in addition to the *PCDH* family of genes, previously reported to be hypomethylated in

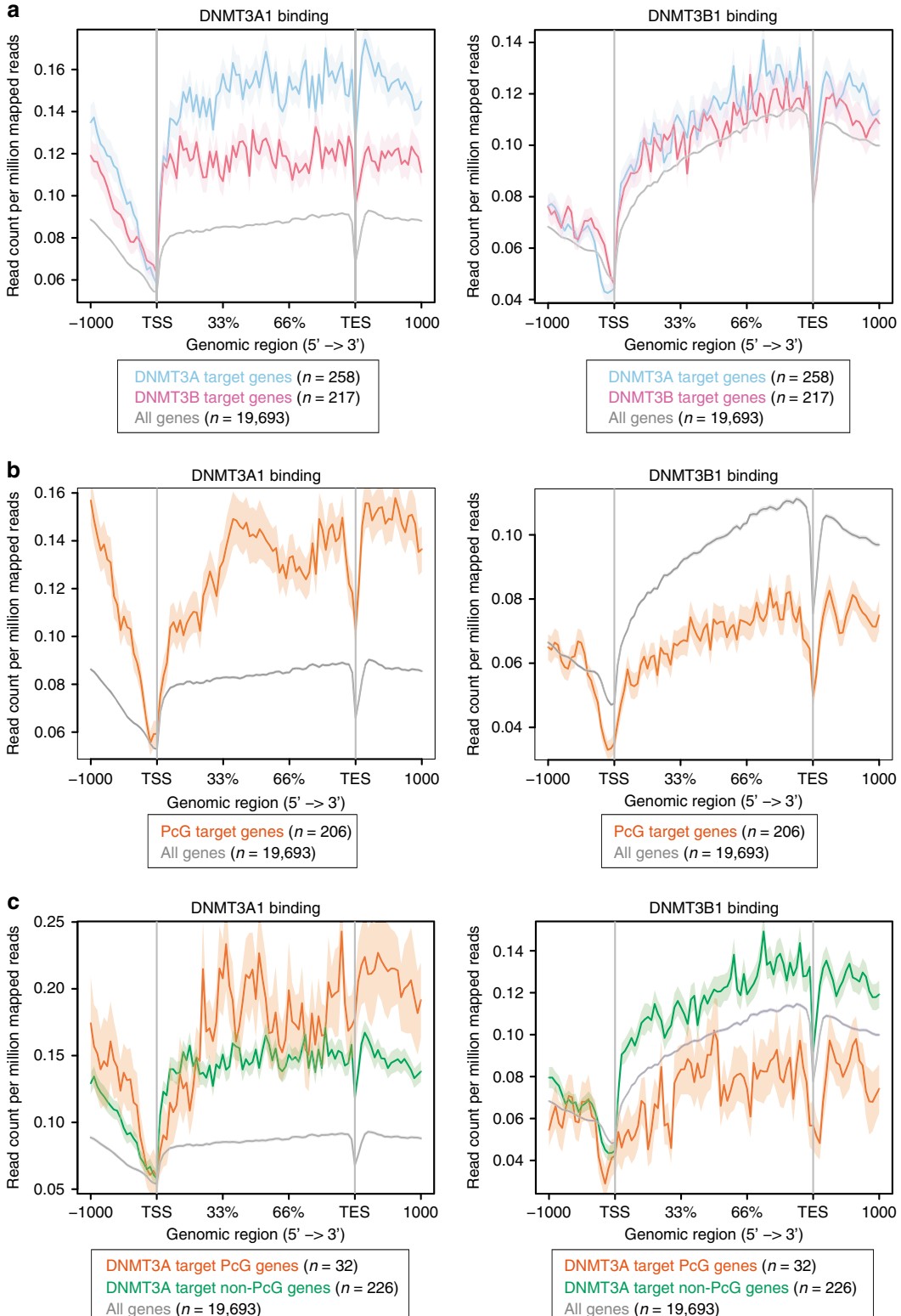

**Fig. 3 Differential binding patterns of DNMT3A1 and DNMT3B1. a** DNMT3A1 and DNMT3B1 binding to target genes for de novo methylation. ChIP-seq data were obtained from GSE57413 and GSE96529[14,15]. Shading represents the standard error of the mean (SEM). **b** DNMT3A1 and DNMT3B1 binding to PcG target developmental genes. Note that DNMT3A1 preferentially binds to PcG target genes, whereas DNMT3B1 is excluded from these genes. Shading represents SEM. **c** DNMT3A1 and DNMT3B1 binding to DNMT3A target genes that are also PcG target genes. Further enrichment of DNMT3A1 was observed on PcG target genes. Note that reduced binding of DNMT3B1 was exclusively detected at PcG target genes among DNMT3A target genes. Shading represents SEM.

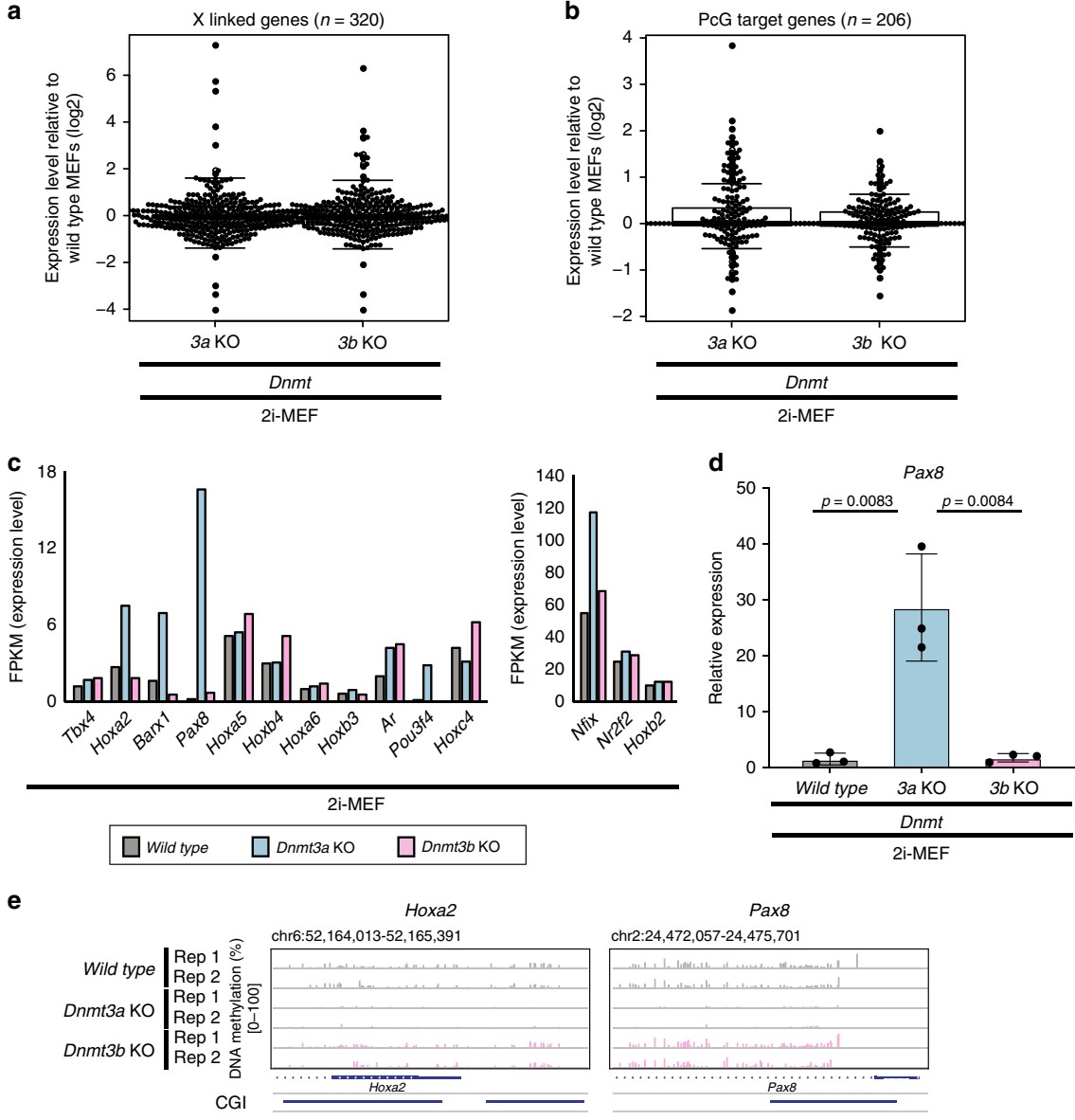

**Fig. 4 Derepression of PcG target genes in Dnmt3a KO 2i-MEFs. a** Relative gene expression levels (FPKM + 1) of X-linked genes (n = 320) in *Dnmt3* KO 2i-MEFs (n = 1) vs. WT 2i-MEFs (n = 1), as determined by RNA-seq. Note that expression levels of most X-linked genes were comparable, regardless of *Dnmt3* status. Solid lines in each box indicate the median. The bottom and top of the boxes are lower and upper quartiles, respectively. Whiskers extend to ±1.5 IQR. **b** Relative gene expression levels (FPKM + 1) of PcG target developmental genes (n = 206) in *Dnmt3* KO 2i-MEFs (n = 1) vs. WT 2i-MEFs (n = 1), as determined by RNA-seq. Note that most PcG target genes were expressed at similar levels. Only a small subset of genes was modestly upregulated in *Dnmt3* KO 2i-MEFs. **c** Expression level (FPKM) of PcG target genes, as determined by RNA-seq. Genes methylated more than 15% at TSS ± 1000 bp in WT 2i-MEFs are shown. A small subset of genes was modestly derepressed in *Dnmt3a* KO 2i-MEFs relative to the other cell types. **d** qRT-PCR analysis for *Pax8* in 2i-MEFs. Note that *Pax8* was derepressed exclusively in *Dnmt3a* KO 2i-MEFs. Data are presented as means ± SD. The mean expression level of 2i-MEFs was defined as 1. Statistical analysis was performed by Student's *t* test (WT vs 3a KO p = 0.0083: WT vs 3b KO p = 0.0084; two-sided, n = 3 biologically independent samples). **e** DNA methylation status at promoters of representative PcG target genes that exhibited modest derepression upon *Dnmt3a* depletion (e.g., *Hoxa2* and *Pax8*). Data from two independent experiments are shown. Each bar indicates a CG site, and bar height represents methylation percentage (0–100%). Locations of genes and CGIs are indicated below.

ICF patients, we identified additional genes that were hypomethylated in both *Dnmt3b* KO 2i-MEFs and ICF patients (e.g., *TMEM151A*) (Fig. 6a,b). Taken together, our results provide candidate genes that may be involved in the pathogenesis of ICF syndrome.

We then examined the relationship between the *DNMT3A* mutations and DNA methylation levels at PcG target developmental genes in AML. Consistent with the assumption that *DNMT3A*-inactivating mutations impair repression of PcG target

developmental genes, previous studies reported that some PcG target developmental genes, such as *HOX* genes and *MEIS1*, are dysregulated in AML and *Dnmt3a* null mouse hematopoietic stem cells[28–30]. To explore this further, we examined a correlation between the presence of the *DNMT3A* mutations and DNA methylation levels at PcG target developmental genes in AML using public datasets (GSE62298). Notably, among 30 differentially methylated PcG target developmental genes, 29 (e.g., *HOXA6*, *HOXB3*, and *MEIS1*) were hypomethylated in *DNMT3A*

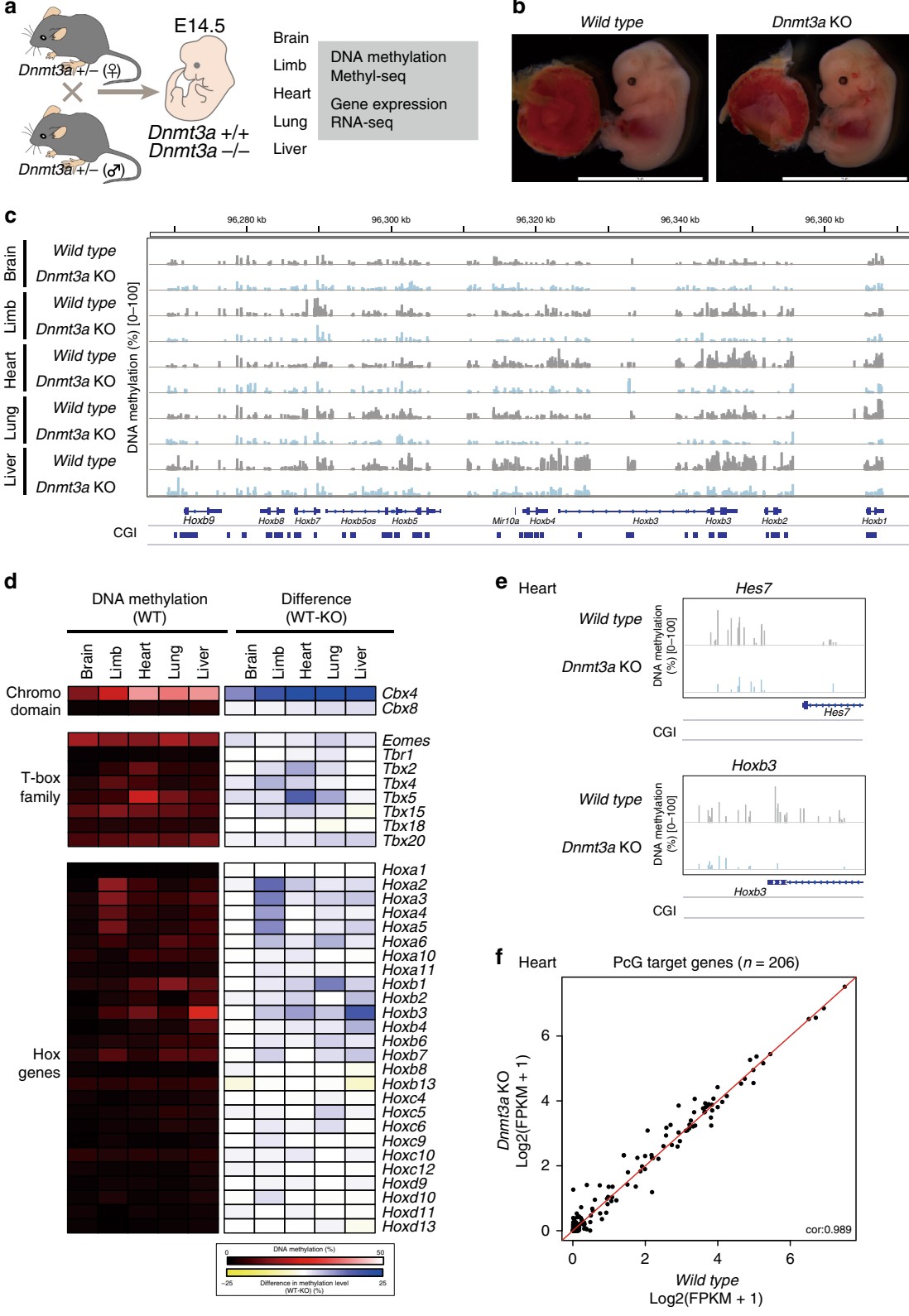

mutant AML (R882: major mutation) relative to *DNMT3A* WT AML, suggesting that inactivating *DNMT3A* mutation causes defects in epigenetic repression of PcG target developmental genes (Fig. 6c, d). These results are also consistent with those of a recent study demonstrating that a gain-of-function mutation at *DNMT3A* causes microcephalic dwarfism and hypermethylation of PcG-regulated genes[31]. Collectively, these data suggest that the region specificity of de novo methylation by DNMT3A and DNMT3B is essentially shared between mice and humans.

## Discussion

Although de novo DNA methylation plays a pivotal role during embryonic development, important unsolved questions include

**Fig. 5 Tissue-specific methylation at PcG target genes by DNMT3A. a** Schematic of experimental design. Various organs (brain, limb, heart, lung, and liver) were obtained from *Dnmt3a* WT and KO embryos at E14.5 for subsequent analyses. **b** Representative image of *Dnmt3a* WT and KO embryos at E14.5. No obvious phenotypic changes were observed in *Dnmt3a* KO embryos. Scale bars, 10 mm. **c** CpG methylation levels at the *Hoxb* cluster in *Dnmt3a* WT and KO organs (brain, limb, heart, lung, and liver), as determined by MethylC-seq. Each bar indicates a CpG site, and bar height represents methylation percentage (0–100%). Locations of genes and CGIs are indicated below. **d** The left panel shows mean DNA methylation levels at CpG sites within each representative PcG target gene in various organs, as determined by MethylC-seq. The right panel shows methylation differences between *Dnmt3a* WT and KO organs. Color scales indicate DNA methylation levels and their differences, respectively. **e** DNA methylation status at promoters of *Hes7* and *Hoxb3*. Each bar indicates a CG site, and bar height represents methylation percentage, as determined by MethylC-seq (0–100%). Locations of genes and CGIs are indicated below. **f** Expression levels (FPKM + 1) of PcG target genes in hearts of *Dnmt3a* WT and KO embryos, as determined by RNA-seq.

precise roles of each DNMT3 enzyme. In this study, we utilized the hypomethylated ES cells as starting materials in a combination with genetic disruption of *Dnmt3* family methyltransferases, and successfully unveiled the distinct region-specificity of de novo methylation activity of these enzymes. Consistent with a previous report, most regions are similarly methylated in WT, *Dnmt3a* KO, and *Dnmt3b* KO cells after differentiation, indicating the redundant function of DNMT3A and DNMT3B. However, we found that DNMT3B has a higher activity of de novo methylation than DNMT3A, which is remarkably pronounced at CGIs on X chromosome. In contrast, de novo methylation at PcG target developmental genes requires DNMT3A, which is different from the mechanism in the extraembryonic lineage. Furthermore, we showed that tissue-specific de novo methylation at PcG target developmental genes is mediated by DNMT3A and is mechanistically associated with stable repression of a small subset of the PcG target genes. Notably, no gene was identified to be hypomethylated in *Dnmt3a* KO epiblasts when compared with WT epiblasts. Indeed, most of identified DNMT3A target genes are unmethylated in epiblasts (Supplementary Fig. 9a, b), suggesting that de novo DNA methylation at these loci occurs after implantation. Unexpectedly, we found that DNMT3A confers gene body methylation at PcG target developmental genes, which is in contrast to other genes being methylated by DNMT3B. Moreover, we showed that human diseases harboring inactivating mutations at *DNMT3* family genes exhibit reduced methylation at hypomethylated regions in mouse *Dnmt3a* and *Dnmt3b* KO cells.

Here we showed that de novo DNA methylation on the X chromosome mainly depends on DNMT3B but not on DNMT3A while de novo methylation at PcG target developmental genes is mediated by DNMT3A. It should be noted that the inactive X chromosome is heavily targeted by Polycomb complexes[32], suggesting that distinct mechanisms are involved in de novo methylation at PcG target genes. Notably, more pronounced reduction in DNA methylation was observed at LINEs in *Dnmt3b* KO than *Dnmt3a* KO 2i-MEFs. Given that greater occurrence of LINEs on the X chromosome than on autosomes, the enriched LINEs may account for the preferential reduction in DNA methylation at the X chromosome in *Dnmt3b* KO 2i-MEFs.

In summary, taking advantage of DNA-hypomethylated 2i/L ES cells, we identified a set of unique de novo target sites for DNA methylation by both DNMT3 enzymes during mammalian development. Our findings have important implications regarding the roles of de novo methyltransferases in epigenetic regulation of mammalian development and diseases.

## Methods

**Data reporting**. No statistical methods were used to predetermine sample size. No experiments were randomized. Investigators were not blinded to allocation during the experiments.

**Establishment and culture of ES cells**. Zygotes with an F1 genetic background (129 × 1/SvJ and MSM/Ms) were obtained by in vitro fertilization. S/L ES and 2i/L ES cells were established in a previous study[16]. Cells were maintained at 37 °C in an atmosphere containing 5% $CO_2$. Briefly, S/L ES cells were established and

maintained on feeders (mitomycin C-treated MEFs) in KO DMEM (GIBCO) containing 2 mM L-glutamine, 100× NEAA, P/S, 15% FBS, 0.11 mM β-mercaptoethanol (GIBCO), and 1000 U/ml human recombinant LIF (Wako). To establish S/L ES cells, a blastocyst was placed on feeders in 24-well plates (passage number 0: p0). After expansion of the ICM, the cells were passaged into 24-well plates (p1), followed by a second passage into six-well plates (p2) and a third passage into 6 cm dishes (p3). For derivation of 2i/L ES cells, a blastocyst was placed on feeders in 24-well plates in KO DMEM containing 2 mM L-glutamine, 100× NEAA, P/S, 15% FBS, 0.11 mM β-mercaptoethanol, 1000 U/ml LIF, 1 μM PD0325901 (STEMGENT), and 3 μM CHIR99021 (STEMGENT) (p0). After passage into 24-well plates (p1), the cells were maintained without feeders in DMEM/F12 (GIBCO) and Neurobasal (GIBCO) supplemented with the B27 (GIBCO) and N2 cell supplement (GIBCO), P/S, 2 mM Glutamax (GIBCO), 1000 U/ml LIF, 1 μM PD0325901, and 3 μM CHIR99021. Cells were passaged into six-well plates (p2) and expanded into 6-cm dishes (p3).

**Genetic deletions of *Dnmt3a* and *Dnmt3b* in ES cells**. Knockout of *Dnmt3a* or *Dnmt3b* for 2i/L ES cells was performed using CRISPR/Cas9[16]. 2i/L ES cells at p3 were used to establish KO lines. sgRNA sequences were as follows: *Dnmt3a*, ACCGCCTCCTGCATGATGCG; *Dnmt3b*: AATGCACTCCTCATACCCGC.

**Establishment of 2i-MEFs by blastocyst injection**. Female ICR mice were treated with pregnant mare serum gonadotropin (PMSG; 7.5 IU) and human chorionic gonadotropin (hCG; 7.5 IU) by intraperitoneal injection. Embryos were rinsed with M2 medium (Sigma) and cultured in KSOM medium until development into blastocysts. WT, *Dnmt3a* KO, and *Dnmt3b* KO 2i/L ES cells were trypsinized, and 7–8 cells per embryo were injected into the blastocoels of E3.5 blastocysts. Injected blastocysts were transferred into the uteri of pseudopregnant ICR mice. A Fully Automated Fluorescence Stereo Microscope Leica M205 FA (Leica MICROSYSTEMS) was used to observe GFP-labeled embryos. MEFs were isolated from E14.5 embryos and cultured in DMEM (Nacalai Tesque) with 2 mM L-glutamine (Nacalai Tesque), 100× nonessential amino acids (NEAA) (Nacalai Tesque), 100 U/ml penicillin, 100 μg/ml streptomycin (P/S) (Nacalai Tesque), and 10% fetal bovine serum (FBS) (GIBCO). To remove host cells, GFP-labeled MEFs were collected by fluorescence activated cell sorting (Aria II, BD) after selection with 350 μg/ml G418 disulfate aqueous solution (Nacalai Tesque) for 7 days.

**Establishment of *Dnmt3a* or *Dnmt3l* WT and KO MEFs**. *Dnmt3a* WT and KO embryos were obtained by crossing *Dnmt3a* heterozygous KO (B6;129S4-Dnmt3a < tm1Enl>) mice[5]. *Dnmt3l* WT and KO embryos were obtained by crossing *Dnmt3l* heterozygous KO mice[12]. MEFs were isolated from E14.5 embryos.

**EpiLC induction**. Culture medium for EpiLCs induction[33] was DMEM/F12 + Neurobasal containing B27 and N2 cell supplements, 20 ng/ml recombinant human activin A (R&D Systems), 12 ng/ml bFGF (Wako), 100 U/ml penicillin, 100 μg/ml streptomycin, and 1% KO serum replacement (GIBCO).

**Animals**. All animal studies were conducted in compliance with ethical regulations in The University of Tokyo and Kyoto University and were approved by Institute of Medical Sciences, The University of Tokyo and Center for iPS Cell Research and Application (CiRA), Kyoto University. MSM/Ms were obtained from RIKEN Bio Resource Center[34,35]. B6;129S4-Dnmt3a < tm1Enl> mice[5] were obtained from RIKEN Bio Resource Center. C57BL/6, 129×1/SvJ, ICR mice, and pseudopregnant ICR mice were purchased from SLC. Noon of the day when a plug was observed was designated as embryonic day (E) 0.5.

**Cell lines**. All cell lines used for this study were derived in our laboratory and tested for mycoplasma contamination.

**Sex determination**. The sex of cell lines and mouse embryos was determined by PCR using primers for *Sry* and *Uty*, both of which are encoded on the Y chromosome. The primers used were as follows: *Sry*-Fw, CGTGGTGAGAGGCACAA

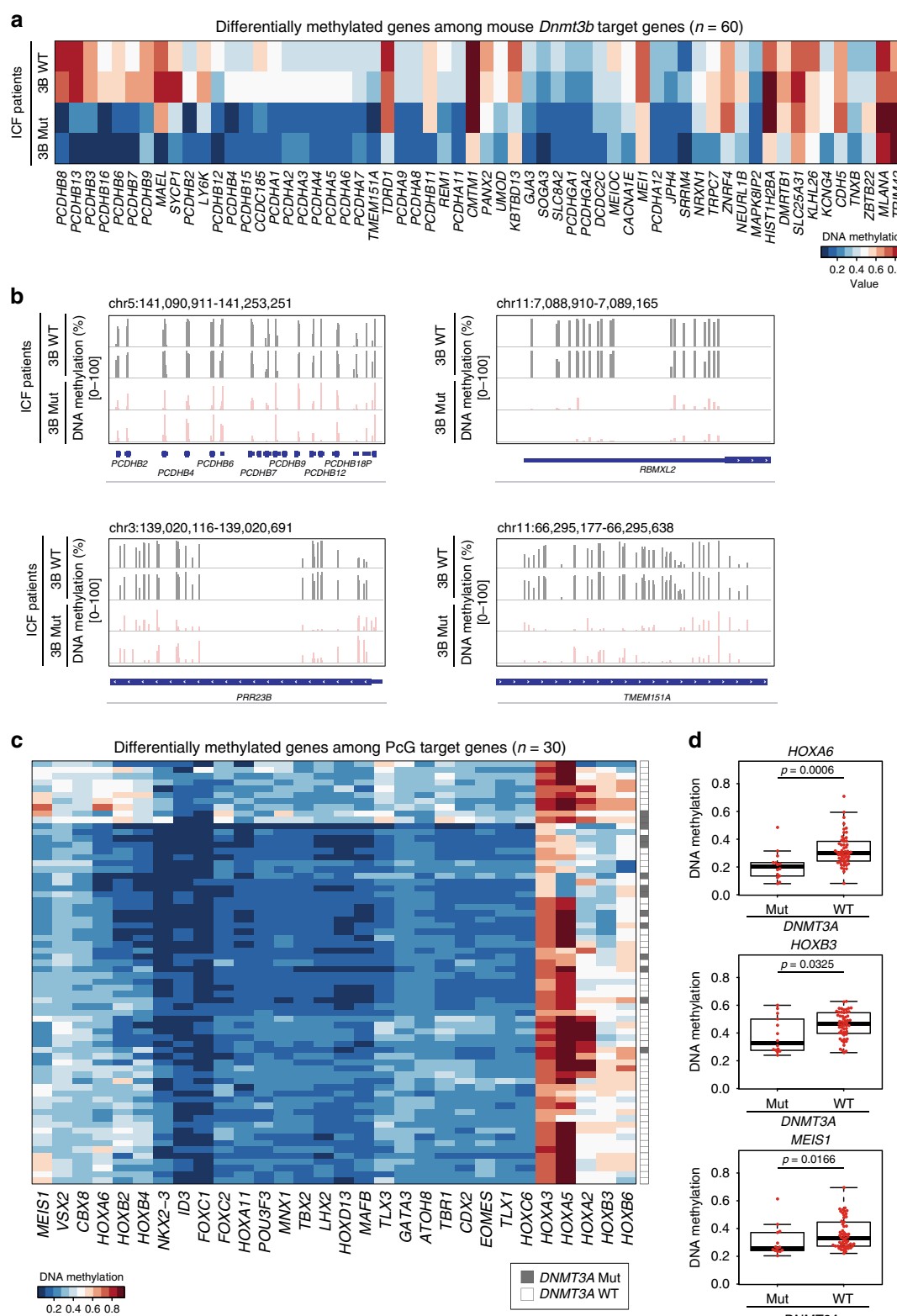

GTT; *Sry*-Rv, AGGCAACTGCAGGCTGTAAA; *Uty*-Fw, GAGTTCTTCTTGCG
TTCACCATCTG; and *Uty*-Rv, CTATCTAATCCACAAAGCGCCTTCTTC.

**GFP labeling of ES cells**. ES cells (p3) were labeled by *piggyBac* transposon carrying *CAG-EGFP-IRES-Neo*[36]. Plasmid and transposase (2.5 μg each) were transfected into ESCs using Xfect mESC Transfection Reagent (Clontech). Transfected cells were selected with 350 μg/ml G418 disulfate aqueous solution. GFP positive colonies were picked and established as GFP-labeled clones (p5).

**DNA/RNA extraction and cDNA synthesis**. DNA was extracted using the PureLink Genomic DNA Mini Kit (Invitrogen), and RNA was extracted using the RNeasy plus Mini Kit (QIAGEN) or NucleoSpin RNA Plus (MACHEREY-NAGEL). DNA and RNA were quantified on a Nanodrop 2000c (Thermo Fisher Scientific) and Qubit (Thermo Fisher Scientific). Reverse transcription was performed using the PrimeScript™ II First strand cDNA Synthesis Kit (Takara).

**Quantitative RT-PCR**. Quantitative PCR was performed using GoTaq qPCR Master Mix and CXR Reference Dye (Promega) on a StepOnePlus Real-Time PCR

**Fig. 6 Reduced DNA methylation in patients with DNMT3 mutations. a** Heatmap of DNA methylation levels at differentially methylated genes in mouse DNMT3B target genes in ICF syndrome patients-derived B cell lines [*DNMT3B* wildtype (WT): two patients, *DNMT3B* mutant (Mut): two patients], as determined by reduced representation bisulfite sequencing (RRBS). Genes differentially methylated (|WT-Mut| >= 0.1) between WT and *DNMT3B* Mut are shown ($n = 60$). Note that most of differentially methylated genes ($n = 57$) exhibited reduced DNA methylation in *DNMT3B*-mutated ICF patients. DNA methylation datasets were obtained from GSE95747[49]. **b** DNA methylation status at representative loci of mouse DNMT3B target genes in ICF patients. DNMT3B target genes in mice were hypomethylated similarly in *DNMT3B*-mutated ICF patients. Each bar indicates a CG site, and bar height represents methylation percentage, as determined by RRBS (0–100%). Locations of genes are indicated below. **c** Heatmap of DNA methylation levels at PcG target developmental genes of 68 AML patients [*DNMT3A* wildtype (WT): 54 patients, *DNMT3A* mutant (Mut): 14 patients] by Infinium assay obtained from GSE62298[29]. Thirty PcG target genes whose average methylation levels within gene bodies differed ($p < 0.05$; two-sided, Mann–Whitney U-test) between *DNMT3A* WT (white) and *DNMT3A* Mut (gray) are shown. Color scale indicates DNA methylation levels. **d** Box plots of DNA methylation levels within gene bodies of *HOXA6*, *HOXB3*, and *MEIS1* in *DNMT3A* WT (54 patients) and *DNMT3A* Mut (14 patients) AML. Bold black lines indicate the median methylation levels. The bottom and top of the box are lower and upper quartiles, respectively. Whiskers extend to ±1.5 IQR. (*HOXA6*, $p = 0.0006$: *HOXB3*, $p = 0.0325$: *MEIS1*, $p = 0.0166$; two-sided, Wilcoxon signed-rank test).

system (Applied Biosystems). Transcript levels were normalized against the corresponding level of β-actin. Primers used were as follows: *Pax8*-Fw, ATGCCTCA CAACTCGATCAGA; *Pax8*-Rv, ACAATGCGTTGACGTACAACTT; *β-Actin*-Fw, GCCAACCGTGAAAAGATGAC; and *β-Actin*-Rv, TCCGGAGTCCATCACAATG.

**Ezh2 inhibitor (GSK126) treatment**. Cells were seeded at a density of $2 \times 10^5$ cells on six-well plates the day before GSK126 treatment. The cells were then cultured in medium containing 10 μM GSK126 (Funakoshi) for 2 days.

**Western blot**. Cells were dissolved in 500 μl of RIPA lysis buffer containing protease inhibitor cocktail (Nacalai Tesque), dithiothreitol (Wako), and phosphatase inhibitor cocktail (Nacalai Tesque). For histone extraction, cells were washed with PBS and treated with trypsin for 5 min. Cells ($1 \times 10^6$) were lysed with extraction buffer on ice. The suspension was centrifuged at 12,000 rpm for 10 s, and the supernatant was removed. Next, 0.2 M $H_2SO_4$ was added to the pellet, and the suspension was then centrifuged at 12,000 rpm for 20 min. The resultant supernatant was mixed with 100% trichloroacetic acid and centrifuged at 12000 rpm for 20 min; the supernatant was removed. Acetone was added to the pellet and lysed. The suspension was then centrifuged at 12,000 rpm for 20 min, and the supernatant was removed. Finally, the pellet was lysed with 1× SDS sample buffer. Proteins were denatured with 2× SDS at 95 °C for 5 min. A total of 30 μg of denatured protein was run on a 10% SDS/PAGE gel and transferred to Amersham Hybond-P PVDF Membrane (GE Healthcare) using PowerPac HC (Bio-Rad). Membranes were blocked in blocking buffer [1× TBS with 0.05% Tween-20 (TBST) containing 4% nonfat milk (Nacalai Tesque)], and then incubated overnight at 4 °C with primary antibodies diluted in blocking buffer. Primary antibodies were as follows: anti-Dnmt3a (Novus Biological, NB120-13888, 64B1446; dilution, 1:500; Santa Cruz Biotechnology, sc-20703; dilution, 1:200) and anti-β-actin (Santa Cruz Biotechnology, sc-47778; dilution, 1:1000). Blots were rinsed with TBST. Membranes were incubated for 90 min at room temperature with HRP-conjugated secondary antibodies. Secondary antibodies used were ECL Anti-mouse IgG, HRP-linked whole antibody from sheep (GE Healthcare, NA931V; dilution, 1:5000), and ECL Anti-rabbit IgG, HRP-linked whole antibody from donkey (GE Healthcare, NA934V; dilution, 1:5000). After rinsing with TBST, Pierce ECL plus Western Blotting Substrate (Thermo Scientific) was used for visualization, and bands were detected on a LAS4000 (GE Healthcare).

**Bisulfite sequencing**. DNA (200 ng) was bisulfite-treated using the EZ DNA Methylation-Gold Kit™ (ZYMO RESEARCH). PCR was performed with EX Taq HS (Takara). PCR products were cloned into pCR4-TOPO vector (Invitrogen). Sub-cloned colonies were sequenced using M13 reverse primer on an ABI 3500xL (Applied Biosystems). Primers were as follows: *Hoxa1*-Fw, AAGTTTGGGG-TAATTTGGATAAAGG; *Hoxa1*-Rv, CCAACCCAAAAAAAAAACCTAAACTAC; *Zfp467*-Fw, AGGTTATTTGTTTGTGTTTAGTGTG; *Zfp467*-Rv, TCCAAATATA AATCTCCCAAATCTATC; *Meis1*-Fw, AGATAGTGAGGGTTTGAAATTTAGT AAA; and *Meis1*-Rv, CCACCCCTAACTCCAAATAAAAAT.

**Library preparation for whole-genome bisulfite sequencing**. DNA (500 ng, as determined by Qubit) was fragmented by sonication (Covaris) and ligated with methylated adapters supplied by TruSeq™ DNA, Sample Prep Kit-v2 (Illumina). Subsequently, DNA was bisulfite-treated using EZ DNA Methylation-Gold Kit™. Final library amplification was performed using *Pfu* Turbo Cx (Agilent Technologies). Sequencing libraries were assessed on a Bioanalyzer (Agilent Technologies) and quantified using the KAPA Library Quantification Kit (KAPA BIOSYSTEMS). The libraries were then sequenced on a HiSeq 2500 (Illumina) as 2× 100 bp or 2× 101 bp paired-end reads.

**Library preparation for target-captured bisulfite sequencing**. DNA (3 μg, as determined by Qubit) was fragmented by sonication. Subsequently, library preparation was performed using the SureSelect^XT Mouse Methyl-Seq Reagent Kit (Agilent Technologies). The Methyl-Seq Kit could enrich 109 Mb of mouse genomic regions including CpG islands, Gencode promoters, DNase I hypersensitive sites, and tissue-specific DMRs. DNA was bisulfite-treated using the EZ DNA Methylation-Gold Kit™. Sequencing libraries were assessed by Bioanalyzer and quantified using the KAPA Library Quantification Kit. The libraries were then sequenced on a HiSeq 2500 (Illumina) as 2× 100 bp or 2× 101 bp paired-end reads.

**Library preparation for RNA sequencing**. RNA (200 ng) was prepared for library construction. High-quality RNA (RNA Integrity Number value ≥8, as determined by Bioanalyzer) was used for library preparation. RNA-seq libraries were generated using the TruSeq Stranded mRNA LT sample prep kit (Illumina). PolyA-containing mRNA was purified by magnetic beads conjugated to poly-T oligos, and RNA was fragmented and primed for cDNA synthesis. Cleaved RNA fragments were reverse transcribed into first-strand cDNA using transcriptase and random primers. Second-strand cDNA was synthesized by incorporation of dUTP, and ds cDNA was separated using AMPure XP beads (Beckman Coulter). A single "A" nucleotide was added to the 3′ ends of the blunt fragments, and adapters with index were ligated to the ends of ds cDNAs. ds cDNA fragments were amplified by PCR using PCR primer Cocktail (Illumina). The number of PCR cycles was minimized (11–15 cycles) to avoid skewing the representation of the libraries. RNA-seq libraries were sequenced on a NextSeq500 (Illumina) as 75 bp single reads.

**DNA methylation data analyses**. For WGBS and MethylC-seq analyses of 129/MSM genetic background cells and tissues, SNP data for MSM/Ms were obtained from the NIG Mouse Genome Database (MSMv4HQ, http://molossinus.lab.nig.ac.jp/msmdb/index.jsp), and the MSM/Ms mouse genome was reconstructed from mm10 using these SNPs. Information about indels was not used in this study. Bases with low-quality scores and the adapters in all sequenced reads were trimmed with cutadapt-1.14[37]. Trimmed reads were mapped independently to both the B6 (mm10) and MSM/Ms mouse genomes using the Bismark software-v0.18.2[38] with bowtie2 (version 2.3.0)[39]. Reads uniquely mapped to the same chromosome and positions of both B6 and MSM/Ms genomes were used for further analyses. B6 (129)-derived and MSM/Ms-derived sequenced reads were determined based on MSM/Ms SNP data. The Y chromosome and mitochondrial genome were omitted from the MSM/Ms reference genome due to the lack of SNP data. SNPs in CpG sites were excluded, and the patterns of bisulfite conversion were taken into account in the determination of parental alleles. Methylated cytosines were extracted from reads using the Bismark methylation extractor with the following options:–ignore 10–ignore_r2 10–ignore_3prime 5–ignore_3prime_r2 5. For MethylC-seq analysis of B6 genetic background tissues, trimmed reads were mapped to the mouse genome (mm10) using Bismark software-v0.18.2 with bowtie2 (version 2.3.0) and default settings, and uniquely mapped reads were used for extraction of methylated cytosines using the Bismark methylation extractor with options–ignore 10–ignore_3prime 5 (single-end reads) or–ignore 10–ignore_r2 10–ignore_3prime 5–ignore_3prime_r2 5. For MethylC-seq analysis, because probe sequences of the SureSelect^XT were designed for the original top strand, only the original bottom data were used for this analysis. The Integrative Genomics Viewer (version 2.8.2)[40] was used to visualize CpG methylated status (CpG sites at ≥5× coverage). Genomic locations for each genetic element were described previously[41]. UCSC LiftOver[42] (http://genome.ucsc.edu/) was used to convert the coordinates of the mm9 assembly to those of the mm10 assembly.

**Identification of differentially methylated loci and regions**. The R/Bioconductor package DSS[43] was used to identify differentially methylated loci and regions. For identification of epiblast-specific methylated loci (Fig. 1b), the DMLtest function in

the DSS package (FDR < 0.05, Epi > ICM) was used with WGBS data (CpG sites at ≥5× coverage) obtained from GSE84236 to determine significantly differentially methylated loci. For identification of DNMT3A and DNMT3B target regions globally (Fig. 2a and Supplementary Data 2), the DMLtest function in the DSS package (FDR < 0.01 and difference in methylation levels (WT–KO) > 0.4) was used with 1k tiled CpG methylation WGBS data (CpG sites at ≥10× coverage) to determine significantly differentially methylated tiles (DMTs; Supplementary Data 1). For identification of DNMT3A and DNMT3B target genes (Fig. 2b, c, Supplementary Fig. 3a, b and Supplementary Data 1–5), the DMLtest and callDMR functions in the DSS package (delta = 0.2, p.threshold = 0.00001, minCG = 10) were used with MethylC-seq data (CpG sites at ≥50× coverage) to determine significantly differentially methylated regions (DMRs, each KO vs WT, Supplementary Data 2 and 4). The genes included in the DMRs (WT > KO) were defined as target genes (Supplementary Data 3 and 5). Gene locations were obtained from the GENCODE annotation file (vM15).

**RNA-seq data analyses**. After trimming of adaptor sequences and low-quality bases with cutadapt-1.15[37], sequenced reads were mapped to the mouse reference genome (mm10) using tophat-2.1.1[44] with the GENCODE version M15 annotation.gtf file and the aligner Bowtie2-2.3.2[39]. Uniquely mapped reads (MAPQ ≥ 20) were used for further analyses. The expression level of each gene was calculated as fragments per kilobase per million mapped reads (FPKM) using cufflinks-2.2.1[45].

**ChIP-seq data analysis**. Raw ChIP-seq data (Dnmt3a and Dnmt3b binding data) in ES cells were obtained from GSE57413[14]. Adaptors and low-quality bases at the 3′ ends of reads were trimmed using cutadapt-1.15[37]. Untrimmed and trimmed reads of 20 bp or more were mapped to the mouse genome (mm10) with BWA 0.7.12[46]. Duplicate reads, chimeric reads, and reads with low mapping quality (MAPQ < 20) were removed using Picard (http://broadinstitute.github.io/picard/) and SAMtools[47]. Reads mapped to blacklisted regions (https://sites.google.com/site/anshulkundaje/projects/blacklists) were excluded from further analysis. Average ChIP-seq plots were drawn using ngsplot-2.47[48].

**Reporting summary**. Further information on research design is available in the Nature Research Life Sciences Reporting Summary linked to this article.

## Data availability

All data analyzed by WGBS, MethylC-seq, and RNA-seq were deposited in the Gene Expression Omnibus (GEO) under accession number GSE111172. MethylC-seq data from *Dnmt3a* WT tissue were deposited under accession number GSE111173. The publicly available datasets used in this study are: GSE84236, GSE95747, GSE62298, GSE57413, GSE96529, GSE95747, GSE62298. All other relevant data supporting the key findings of this study are available within the article and its Supplementary Information files or from the corresponding authors upon reasonable request. The source data underlying Supplementary Fig 4c are provided as a Source Data file. A reporting summary for this Article is available as a Supplementary Information file.

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

## Acknowledgements

We are grateful to S. Sakurai and D. Seki for technical assistance. Y. Y. was supported in part by P-CREATE (JP18cm0106203h0003), Japan Agency for Medical Research and Development (AMED); AMED-CREST (JP19gm1110004h9903), AMED; JSPS KAKENHI (18H04026, 20H05384); the Princess Takamatsu Cancer Research Fund; the Takeda Science Foundation; the Mochida Foundation; and the Naito Foundation. T.Y. was supported by AMED-CREST (19gm1110004s0203); JSPS KAKENHI 19H03418; Core Center for iPS Cell Research, Research Center Network for Realization of Regenerative Medicine, AMED; and the iPS Cell Research Fund. M. Y. was supported by JSPS KAKENHI 15J05792. The ASHBi is supported by World Premier International Research Center Initiative (WPI), MEXT, Japan.

## Author contributions

M.Y., T.Y., and Y.Y. designed and conceived the study and wrote the manuscript. M.Y. and T.U. generated cell lines, performed experiments, and generated WGBS and Methyl-seq libraries. A.T. performed blastocyst injections. K.N. and K.H. provided *Dnmt3l* WT and KO MEFs. M.K., S.O., M.S., A.M., and T.Y. analyzed ChIP-seq, WGBS, Methyl-seq, and RNA-seq data in mouse and human.

## Competing interests

The authors declare no competing interests.
