## [Peer Review File · Nature Communications]

Reviewers' comments:

Reviewer #1 (Remarks to the Author):

In this manuscript Yagi et al describe DNMT3A/B targeting preferences during mammalian development. Towards this they utilise genetic ablations of DNMT3A and DNMT3B in 2i ES cells and derived fibroblasts. They identify unique sites that are methylated by either DNMT3A or DNMT3B. In particular, they identify Polycomb genes as targets of DNMT3A.

The authors make use of a smart approach where they employ hypomethylated 2i ES cells to mimic the ICM DNA methylation state. They use these ES cells to derive MEFs lacking either Dnmt3a or Dnmt3b and perform WGBS to obtain Dnmt3-specific methylation profiles and gene expression.

This work provides a good resource on the de-novo DNA methylation targets during development, but little additional insights about mechanisms and relevance of the observed targeting specificity. That said, the study is well described and the experiments and presented data generally support the conclusions made in this manuscript.

I have a few comments that need to be considered by the authors:

1) 10k tiles are used to detect DMRs, I am worried that this could lead to misinterpretation of the results. In particular the number and position of the called DMRs could be affected. I strongly recommend that the authors repeat DMR identification using smaller tiles (e.g. 1kb) or dynamic DMR detection (i.e. without predefined tile size).

2) The 10kb tiles can furthermore influence the association of DMRs with target genes, GO, etc.... This analysis should be also repeated with an alternative DMR detection approach.

3) The definition of the gene-body region is problematic. 1000bp downstream from TSS is a too short distance and can still contain considerable parts of regulatory regions of the gene promoters (including CpG island shores). Here the authors should re-define the gene bodies starting at 3kb downstream of the TSS. Furthermore it is not clear if in their analysis the authors removed gene bodies (or parts of gene bodies) that overlap with promoters of other genes, or alternative downstream promoters.

4) Related to the point above, gene-size can strongly influence the distinction between promoter and gene bodies. Short genes (e.g. smaller than 3-4kb) tend to have promoter chromatin features extending throughout the gene body. This is particular important for their conclusion that Dnmt3a regulates gene bodies of Polycomb target genes. Polycomb target genes are usually very short genes that consist of a few exons with short introns (e.g. Hox genes and other TFs). In these cases, the entire gene body is largely covered by K27me3/H3K4me3 and is hypomethylated. Here the authors need to distinguish between short and long genes. They need to prove that after excluding all short genes they can still observe gene body methylation by DNMT3a. Otherwise, the observed changes are rather resulting from changes that happen adjacent to CpG island promoters (e.g.shores), but not at gene bodies.

Reviewer #2 (Remarks to the Author):

DNA methylation plays critical roles in embryo development as well as human disease progression. DNMTs are responsible for DNA methylation establishment and maintenance, and to identify their target sites during embryogenesis implicates the distinct functions of do novo methyltransferases in epigenetic regulation of embryo development as well as human diseases. In this study, Yagi and

his/her colleagues took advantage of the model of ES cells with genetic ablations of dnmt3 methyltransferases, and identified distinct target loci of the de novo methylation activity of dnmt3a and dnmt3b, respectively. Subsequently, the authors found some similar loci with reduced DNA methylation level in human patients harboring DNMT3A and DNMT3B somatic mutations, suggesting shared mechanism of de novo DNA methylation in mouse and human. The whole study was well designed and data was interpreted carefully, observations were likely to generate considerable interest in the field. However, some key technical as well as conceptual challenges (see below) need to be fully addressed before I make further decisions.

1. There is a drastic genome-wide remethylation during embryo implantation, followed by considerable methylation changes during the organogenesis and both dnmt3a and dnmt3b are involved in these processes. Throughout the manuscript, authors did most of the comparisons between dnmt3 KO MEFs and WT MEFs and identified some loci specific to dnmt3a (or dnmt3b); it would be better if authors could first compare the methylation difference between dnmt3 KO epiblasts and WT epiblasts to identify one set of loci specific to either dnmt3a or dnmt3b during implantation.
2. The authors did most of the sequencing experiments without replicates, and most of the comparisons were carried out without any statistical corrections. It would be necessary to sequence at least one more batch to see if the findings are reproducible or not.
3. It would be great to see if there are some consensus DNA sequences/TF binding sites within dnmt3a and dnmt3b specific loci.
4. Bisulfite conversion rate should be shown as one of the sequencing QC metrics, and also the methylation status of the imprinting loci in KO-MEFs and KO-tissues should be shown and discussed.
5. Bisulfite treatment is indistinguishable of DNA methylation and DNA hydroxymethylation, all the patterns authors observed are the average level of DNA methylation plus DNA hydroxymethylation, authors should discuss this point.
6. Page11, line 277, the sentence is truncated.

Yagi et al., “Genome-wide identification of distinct target loci for *de novo* DNA methylation by DNMT3A and DNMT3B during mammalian development”

Response to referee’s comments:

We thank the reviewers for their helpful comments and suggestions. We have performed additional experiments to address their questions and concerns. We believe that our manuscript has been substantially improved as a consequence. Particularly, we performed WGBS and MethylC-seq using different samples of 2i-MEFs and re-analyzed most of DNA methylation data, which strengthened our conclusions. We have responded to each reviewer’s point individually in the subsequent section. We hope that these experiments and our responses will clarify all concerns and that our manuscript is now suitable for publication in *Nature Communications*.

Response to referee's comments:

Reviewer #1 (Remarks to the Author):

In this manuscript Yagi et al describe DNMT3A/B targeting preferences during mammalian development. Towards this they utilise genetic ablations of DNMT3A and DNMT3B in 2i ES cells and derived fibroblasts. They identify unique sites that are methylated by either DNMT3A or DNMT3B. In particular, they identify Polycomb genes as targets of DNMT3A.

The authors make use of a smart approach where they employ hypomethylated 2i ES cells to mimic the ICM DNA methylation state. They use these ES cells to derive MEFs lacking either Dnmt3a or Dnmt3b and perform WGBS to obtain Dnmt3-specific methylation profiles and gene expression.

This work provides a good resource on the de-novo DNA methylation targets during development, but little additional insights about mechanisms and relevance of the observed targeting specificity. That said, the study is well described and the experiments and presented data generally support the conclusions made in this manuscript.

We thank the reviewer for his/her enthusiasm.

I have a few comments that need to be considered by the authors:

1) 10k tiles are used to detect DMRs, I am worried that this could lead to misinterpretation of the results. In particular the number and position of the called DMRs could be affected. I strongly recommend that the authors repeat DMR identification using smaller tiles (e.g. 1kb) or dynamic DMR detection (i.e. without predefined tile size).

We apologize for not clearly describing the methods to identify target regions and target genes of DNMT3A and DNMT3B. In our previous manuscript, two different methods were used to identify the target regions (10kb tiles, global detection) and target genes (dynamic DMR detection, local detection). In agreement with the reviewer's comments that smaller tiles are better, we used 1kb tiles instead 10kb tiles with WGBS data for

global identification of DMRs (Figure 2a and Table S1) by applying the Bioconductor DSS package (Figure for reviewers, upper panels). Moreover, by using a dynamic DMR detection method in the Bioconductor DSS package (Figure for reviewers, lower panels), uniquely hypomethylated genes in *Dnmt3* KO 2i-MEFs are identified with MethylC-seq data that allow us to examine the target gene regions in more detail. These procedures are clearly described in the Method section of our revised manuscript.

As suggested by this reviewer, we used 1kb tile instead of 10kb tile for genome-wide identification of differentially methylated loci with WGBS data (Figure 2a and Table S1). We used dynamic DMR detection to identify uniquely hypomethylated genes in *Dnmt3a* and *Dnmt3b* KO 2i-MEFs. The detailed protocol for the dynamic DMR detection is described in the Methods section.

2) The 10kb tiles can furthermore influence the association of DMRs with target genes, GO, etc.... This analysis should be also repeated with an alternative DMR detection approach.

As described above, we used dynamic DMR detection to identify uniquely unmethylated genes in *Dnmt3a* and *Dnmt3b* KO 2i-MEFs.

3) The definition of the gene-body region is problematic. 1000bp downstream from TSS is a too short distance and can still contain considerable parts of regulatory regions of the gene promoters (including CpG island shores). Here the authors should re-define the gene bodies starting at 3kb downstream of the TSS. Furthermore it is not clear if in their analysis the authors removed gene bodies (or parts of gene bodies) that overlap with promoters of other genes, or alternative downstream promoters.

We thank this reviewer for thoughtful and helpful comments. We agree that Polycomb target developmental genes are usually short genes, thus it is possible that the observed changes are resulting from changes that happen adjacent to promoters. Indeed, we found that a DNA hypomethylated region in *Gata3* was preferentially observed adjacent to a CpG island near the TSS. As suggested by this reviewer, we re-defined the gene-bodies (starting at 3kb downstream of the TSS to TES) and analyzed

gene-body DNA methylation at PRC target developmental genes (Figure S4a and S5b). We confirmed the gene-body DNA hypomethylation in *Dnmt3a* KO 2i-MEFs even with the definition. We did not exclude genes that overlap with promoters of other genes and alternative downstream promoters in this analysis. Still, considering that gene-bodies of longer genes similarly exhibit *Dnmt3a*-mediated DNA methylation (see below, Figure S4a and S5b), we concluded that the observed changes at the gene-bodies are not resulting from close proximity of CpG islands linked to promoters.

4) Related to the point above, gene-size can strongly influence the distinction between promoter and gene bodies. Short genes (e.g. smaller than 3-4kb) tend to have promoter chromatin features extending throughout the gene body. This is particular important for their conclusion that *Dnmt3a* regulates gene bodies of Polycomb target genes. Polycomb target genes are usually very short genes that consist of a few exons with short introns (e.g. Hox genes and other TFs). In these cases, the entire gene body is largely covered by K27me3/H3K4me3 and is hypomethylated. Here the authors need to distinguish between short and long genes. They need to prove that after excluding all short genes they can still observe gene body methylation by DNMT3a. Otherwise, the observed changes are rather resulting from changes that happen adjacent to CpG island promoters (e.g.shores), but not at gene bodies.

We divided genes into short (3-5 kb) and long genes (>5 kb), and analyzed DNA methylation levels at PRC target developmental genes (Figure S4a and S5b). We observed reduced DNA methylation levels in both short and long genes in *Dnmt3a* KO 2i-MEFs (Figure S4a and S5b), affirming that *Dnmt3a* regulates DNA methylation at gene-bodies of Polycomb target genes.

Reviewer #2 (Remarks to the Author):

DNA methylation plays critical roles in embryo development as well as human disease progression. DNMTs are responsible for DNA methylation establishment and maintenance, and to identify their target sites during embryogenesis implicates the distinct functions of *de novo* methyltransferases in epigenetic regulation of embryo development as well as human diseases. In this study, Yagi and his/her colleagues took advantage of the model of ES cells with genetic ablations of *dnmt3* methyltransferases, and identified distinct target loci of the *de novo* methylation activity of *dnmt3a* and *dnmt3b*, respectively. Subsequently, the authors found some similar loci with reduced DNA methylation level in human patients harboring DNMT3A and DNMT3B somatic mutations, suggesting shared mechanism of *de novo* DNA methylation in mouse and human. The whole study was well designed and data was interpreted carefully, observations were likely to generate considerable interest in the field. However, some key technical as well as conceptual challenges (see below) need to be fully addressed before I make further decisions.

We thank the reviewer for his/her enthusiasm.

1. There is a drastic genome-wide remethylation during embryo implantation, followed by considerable methylation changes during the organogenesis and both *dnmt3a* and *dnmt3b* are involved in these processes. Throughout the manuscript, authors did most of the comparisons between *dnmt3* KO MEFs and WT MEFs and identified some loci specific to *dnmt3a* (or *dnmt3b*); it would be better if authors could first compare the methylation difference between *dnmt3* KO epiblasts and WT epiblasts to identify one set of loci specific to either *dnmt3a* or *dnmt3b* during implantation.

As suggested by the reviewer, we compared DNA methylation between *Dnmt3* KO epiblasts and WT epiblasts. Importantly, we did not identify any gene that is hypomethylated in *Dnmt3a* KO epiblasts when compared with WT epiblasts, suggesting that DNMT3A does not play a significant role on *de novo* DNA methylation during peri-implantation. Consistent with this, we found that PRC target developmental genes are generally unmethylated in epiblasts (Figure S9a and S9b), indicating that *de novo*

methylation at PRC target developmental genes occurs during the organogenesis. We also found that *Dnmt3b* KO epiblasts exhibit reduced methylation at CpG sites that acquire *de novo* DNA methylation during peri-implantation (from ICM to epiblasts) (Figure S2b). These findings support our conclusions that *de novo* DNA methylation activity of DNMT3B is predominant in early mouse development. We added these results and discussions in the revised manuscript.

2. The authors did most of the sequencing experiments without replicates, and most of the comparisons were carried out without any statistical corrections. It would be necessary to sequence at least one more batch to see if the findings are reproducible or not.

We appreciate this reviewer's important suggestions. In the revised manuscript, we performed WGBS and MethylC-seq of different samples of 2i-MEFs. Accordingly, we re-analyzed data from two different experiments and revised the related Figures throughout the manuscript (Figure 1, 2, 3, 6, S1, S2, S3, S4, S5, S6). We were able to reproduce results in the previous version, which strengthened our conclusions. We also performed conventional bisulfite sequencing and confirmed the reduced DNA methylation levels at PRC target genes in various organs of *Dnmt3a* KO embryos (Figure S8a). We also added statistical analyses in Figure 1i, 2d, 2e, 4a, 4b, 6d, S3e, S4a, S5b, and S6b.

3. It would be great to see if there are some consensus DNA sequences/TF binding sites within *dnmt3a* and *dnmt3b* specific loci.

We thank this reviewer for thoughtful comments. We have performed motif analysis for DNMT3A-/DNMT3B-specific loci using a HOMER software. However, no specific motif was enriched more than two times in both loci when compared with the background sequences, suggesting that specific transcription factors do not have a major impact on the locus specificity of each locus (data not shown).

4. Bisulfite conversion rate should be shown as one of the sequencing QC metrics, and also the methylation status of the imprinting loci in KO-MEFs and KO-tissues should be shown and discussed.

In the revised manuscript, we included the bisulfite conversion rate in Figure S1b. We also showed loss of genomic imprinting in 2i-MEFs and discussed this in the text (Figure S1d).

5. Bisulfite treatment is indistinguishable of DNA methylation and DNA hydroxymethylation, all the patterns authors observed are the average level of DNA methylation plus DNA hydroxymethylation, authors should discuss this point.

We appreciate this reviewer's important suggestion. 5mC cannot be distinguished from 5hmC by bisulfite sequencing (Hayatsu and Shiragami, 1979). Given that the 5hmC level is negligible in ESCs (Shirane et al. 2016), we describe both modified bases as 5mC in the present study. We have mentioned this point in the text (Results section).

6. Page11, line 277, the sentence is truncated.

We apologize for this mistake and have corrected it.

Figure for reviewers

Figure 2a and TableS1
(Differentially methylated tile (1K), WGBS)

Figures 2b, 2c, S3a and S3b and Table 1~4
(Dnmt3A and Dnmt3b target genes after Figure 2b, MethylC-seq)

Differentially methylated loci (DML)

(DMLtest function in the DSS package)

Differentially methylated region (DMR)

(callDMR function in the DSS package)

DNMT3s Target genes

REVIEWERS' COMMENTS:

Reviewer #1 (Remarks to the Author):

The authors have addressed all my requests.

Reviewer #2 (Remarks to the Author):

The authors have addressed my concerns to a satisfactory extent, I recommend for publication.

Response to REVIEWERS' COMMENTS:

Reviewer #1 (Remarks to the Author):

The authors have addressed all my requests.

Reviewer #2 (Remarks to the Author):

The authors have addressed my concerns to a satisfactory extent, I recommend for publication.

We thank both reviewers for the constructive comments during the review process.